# Perceptions of Cheating and Doping in E-Cycling

**DOI:** 10.3390/sports11100201

**Published:** 2023-10-14

**Authors:** Andrew Richardson, Nicolas Berger, Phillip Smith

**Affiliations:** 1School of Social Sciences, Humanities and Law, Teesside University, Middlesbrough TS1 3BX, UK; 2School of Health & Life Sciences, Teesside University, Middlesbrough TS1 3BX, UK; n.berger@tees.ac.uk (N.B.); phillip.smith@tees.ac.uk (P.S.)

**Keywords:** E-cycling, doping, anti-doping, Zwift, and cheating

## Abstract

E-cycling is a growing area of cycling appealing to competitive cyclists and fitness enthusiasts. Zwift is the most popular e-cycling platform, with approx. 1 million subscribers and is a virtual environment that hosts regular races, including the UCI e-cycling world championships. The popularity of Zwift has given rise to cases of cheating and hacking the system to gain an advantage in e-racing. As a result, some high-profile professional riders have faced bans. We set out to understand the thoughts and concerns e-cyclists have about cheating, hacking, and doping in e-cycling. A total of 337 females and 1130 males were recruited over a 7-week period via social networking sites to complete an online survey. Forty-four per cent had experienced cheating during e-racing, which made them feel angry, annoyed, disappointed, and cheated. However, 15% of those who experienced cheating said they did not care, possibly because many see e-racing as a game or training tool rather than a competitive event. Eighty-seven per cent of participants were in favour of enforcing a ban on cheaters in e-cycling, while 34% wanted cheaters to be banned during in-person cycling events too. Results indicate that many e-cyclists have experienced cheating and would like clearer rules and bans for cheaters during e-races.

## 1. Introduction

The field of electronic sports or “esports” is now a well established part of the gaming and entertainment industries [1,2]. Since Stanford University students competed in Spacewar, the first competitive esports event in 1972 [3], esports has seen rapid expansion [4]. Jenny et al. [5] state that esports can have many definitions based on the different games and genres. Popular esports games can be based on sci-fi, historical or political events, or fantasy [6,7,8]. Esports have dedicated leagues, tournaments, and events for each game. Some popular and well-known esports games include Counter-Strike, DOTA 2, League of Legends, StarCraft 2, Fortnite and Apex Legends [9]. These games are categorised as traditional first-person shooters (FPS), multiplayer online battle arenas (MOBA) and battle royale games. Esports can also replicate existing sports, such as EA Sports covering NBA, Madden NFL and FIFA replicating basketball, American football, and soccer, respectively, and Forza replicating motor racing [9]. These games that emulate real-life sports but do not involve physical activity are known as “sedentary sport video games” (SSVG) [10]. In this paper, we define esports as “motion-based video games (MBVG) that track gross motor physical body movements within the game” [5], such as Zwift and other cycling turbo-trainer-based platforms. Irrespective of the different types of esports games and platforms out there, they all attract a diverse range and thousands of viewers.

Esports leagues and tournaments attract hundreds of thousands of spectators both online and in person [11]. Recent esports events are predicted to exceed previous viewership records and challenge the spectatorship of traditional sports. However, it is currently some way off the fanbase and engagement of the largest sporting events, such as the FIFA World Cup Finals and The Olympic Games (see Table 1 for comparison) [12]. Due to the large fanbase esports has attracted, there is a significant prize pool that competes with some of the largest sporting events. For example, in 2021, the winning team in a DOTA 2 tournament won 18 million USD [13,14].

Focusing on esports games that require physical activity, these MBVGs have made significant strides towards the popularity of in-person sports [17,18]. Such sports include e-rowing events, which have their own racing leagues for the Concept2 Rower [19,20] and virtual golf simulators, which function as both a training tool and social entertainment platform [21,22]. Finally, the focus of this paper will investigate Zwift and other cycling-based platforms for e-racing and training [17,18].

## 2. E-Cycling

Cycling MBVGs can be traced back to an advert from 1888 promoting the use of indoor cycling rollers combined with a cinematograph and electric fan to simulate riding outdoors [23]. The 2010s saw the release of the latest generation of cycling “smart trainers” which connect to a road bike with a roller against the rear tyre, or in place of the rear wheel, to electronically control riding resistance and provide connectivity with cycling MBVG platforms. 

E-cycling platforms vary in design from training-focused platforms like TrainerRoad and Wahoo System to those featuring videos of real-life riders such as FulGaz or hybrids of virtual reality and real-life like BKool and Rouvy, to fully simulated platforms like Road Grand Tours (RGT) and Zwift, which both operate in a virtual world [24]. RGT focuses on simulating real-life routes, both pre-rendered in the application, and their “magic routes” feature allows users to upload a file with an existing route and have a rideable version in their virtual world [25]. The most common platform, Zwift, includes animated versions of real-life roads/routes, and those are based entirely on fantasy. Riders are represented by an on-screen avatar based on their physical proportions (height/mass). This affects in-game physics to simulate real-life cycling dynamics, such as uphill riding and aerodynamics, when riding solo and drafting other riders [26]. In a recent interview, Co-CEO Eric Min stated that Zwift has around 1 million subscribers [27]. 

Zwift received validation as the preeminent platform for e-cycling when it was chosen by cycling governing bodies to host their championships, such as the British Cycling e-cycling championships in 2019 and the Union Cycliste Internationale (UCI) World e-cycling Championships in 2020. Since its Beta launch in 2014, Zwift has grown with the addition of new virtual worlds and was significantly aided by the COVID-19 pandemic lockdowns in Spring 2020 [28], turning riders in the northern hemisphere, which would traditionally be looking to increase the amount of outdoor riding, to an indoor alternative [29,30] to follow government policies. 

To fully understand the issues facing e-cycling, we will outline cycling and its history of cheating and doping before investigating how e-cycling addresses these issues.

## 3. Cheating and (Anti) Doping in Cycling

Cheating and doping have unfortunately been a part of cycling since the earliest cycling competitions. In 1896, Arthur Linton reportedly died of exhaustion and typhoid fever, although it was rumoured that he died as a result of being doped with strychnine by his coach [31]. Drugs and cycling have been intertwined to the extent that the race director for the 1930 Tour de France had to specify that he would not be supplying drugs for the riders, so they would need to bring their own. Famous cyclists such as Fausto Coppi in the 1940s and Jacques Anquetil in the 1960s admitted to taking amphetamines to boost performance [32]. In 1960, the first anti-doping law was passed in France, and the first anti-doping test was at the Tokyo Olympic Games. This was followed by testing from the UCI in 1966 and the Tour de France in 1967.

Anti-doping testing progressed from direct testing for drugs such as amphetamines and anabolic steroids in the 1970s [33] to the banning of blood transfusions in the 1980s, although no test was available to detect this until much later. Prior to the 1970s, all testing was only conducted in competition until the Norwegian authorities began performing out-of-competition (OOC) tests in 1977 [33]. The World Rowing Governing Body (FISA) was the first governing body to announce OOC testing in 1982, and by the 1990s, this was conducted by most sports [34]. Unannounced OOC testing became the norm in the 1990s as they adopted the World Athletics Governing Bodies (the IAAF) anti-doping flying squad, which was designed to go anywhere and test anyone [33]. This was mostly limited to non-competition days during national and world championships [34], as the OOC testing system of the 1990s was so inefficient that testers were completing fewer than half of their testing missions because of athlete unavailability. In the early 2000s, the United States Anti-Doping Agency (USADA) was the first to implement a whereabouts system and the three-strikes rule, under which missing three tests was a sanctionable offence [33]. No one fell afoul of this rule in cycling until Michael Rasmussen in 2006. The following year [35], Rasmussen was ejected from the Tour de France by his own team whilst leading the race due to lying about his whereabouts before the race started.

Doping is defined by the World Anti-Doping Agency (WADA) as “the occurrence of one or more of the anti-doping rule violations”. In our previous work [36], we summarise the multiple anti-doping rule violations (ADRVs) that are classified as doping or cheating. 

The current total number of cyclists being tested is challenging to acquire, especially before and after the pandemic (see Table 2); how many active riders exist and how many tests are conducted in total are readily available data. However, how many repeated tests are performed on the same riders is difficult to determine. In-competition testing partly depends on the finishing position, as those placing higher are more likely to be tested.

Cheating in cycling was commonplace, with notable examples including taking trains and being towed by a car or motorbike while holding a cork attached to a wire in the mouth [38]. There have been cases of people throwing tacks on the road to cause punctures and poisoning the drinks of rivals [38,39]. A grey area still exists around riders holding on to cars in the race convoy while coming back from a crash or receiving medical or mechanical support. This is tolerated by race officials to an extent but can result in disqualification when done excessively, on a climb, or to gain an advantage [39]. Drafting behind cars is also not officially allowed, but in practice, it is widely accepted when returning to the main group from a crash. In more severe instances, many riders have been penalized with a fine, time addition in stage races, or disqualification. Vincenzo Nibali (a former winner of the Tour de France and Giro d’Italia) was famously disqualified from the Vuelta a España in 2015 when TV cameras filmed him holding on to his team car while it accelerated away from a pack of other riders to rejoin the front group he had been dropped from [40].

Serious allegations of mechanical doping with hidden motors installed in bikes were first made after the 2010 Tour of Flanders when Fabian Cancellara attacked a steep climb while remaining seated, which was unusual given the steep gradient. Allegations of a hidden motor in his bike were also made against Ryder Hesjedal when a video of him crashing during the 2014 Vuelta a España emerged, and his rear wheel kept spinning while the bike was on the ground [41]. Although it was likely due to the momentum of the wheel at the time of the crash, suspicions continued around hidden motors until the first confirmed case in January 2016, when Femke Van den Driessche was found to have a motor hidden in her bike during the U23 UCI Cyclocross World Championships. For this offence, she was given a 6-year ban and a EUR 20,000 fine [42]. However, this is, to date, the only case of detected motor doping in professional cycling. 

The level of cheating and doping in cycling has been well represented in the scientific literature [36] and the mainstream media [43,44]. The most famous and well-known doping case is that of Lance Armstrong and the US Postal Team (later Discovery Team). Armstrong was stripped of his seven Tour de France titles for doping throughout most of his career, and at the time of writing, no winner is listed for the 1999–2005 Tour de France. Post-Armstrong, cycling still battles doping and cheating cases in traditional and e-cycling events [37]. Although the number of those caught doping has decreased, it is still to be determined whether this is due to fewer athletes doping or due to newer and more difficult-to-detect drugs and/or micro-dosing regimens [45].

## 4. Doping and Cheating in Esports and E-Cycling

Doping and taking performance-enhancing drugs (PEDs) is not uncommon in esports. For example, in a live-streamed esports competition, a player openly admitted to using Adderall [46]. Schubert, Eling and Konecke [47] interviewed nine professional FIFA esports players to discuss their perceptions of the use of drugs and other (alleged) PEDs. The authors found that the three main themes that developed included high-performance pressure to perform in these events, an ambivalent perception towards the different forms of PEDs to supplement gaming ability, and the apparent lack of any anti-doping systems or measures. One respondent argued that the definition of doping in FIFA esports is problematic because it cannot be compared to physical exertion in “traditional” or in-person football [47]. Many of the respondents welcome some form of anti-doping enforcement to their sport. However, the difficulty lies in controlling, monitoring, and catching cheaters and dopers when they are spread across the globe rather than on a football pitch.

Friehs et al. [48] drew parallels with other cycling anti-doping research, as many athletes who feel they must dope do so because they believe other athletes are already doping. This can be seen in Lentillon-Kaester, Hagger and Hardcastle [49], which looked at health and doping in 16 elite-level cyclists in 2012. They found that perceived health hazards did not influence the athletes’ decisions to take PEDs most of the time and a trivialization of the side effects of using PEDs. Finally, the younger cyclists in the sample were not concerned with the risks of the long-term side effects of these substances and focused on what could be gained in the short term. 

Westmattelmann et al. [50] investigated perceptions of anti-doping regimes among 42 top-level German cyclists and 104 track and field athletes. Here, the cyclists attested to the importance of follow-up testing for effective anti-doping policies. They also noted the effectiveness of the athlete biological passport (ABP) due to it being introduced into cycling first and catching some high-profile riders [51]. However, the cyclists perceive the effectiveness of the increased use of the Anti-Doping Administration and Management System (ADAMS) as very low due to the time-consuming admin of logging whereabouts, privacy issues regarding how the data is stored and who has access to it, and how it is used, which may lead to mistrust in the system.

Since the high-profile case of cheating in the 2019 British e-cycling national championship, which resulted in a rider being banned from Zwift and in-person events [52], there has been a surge in research examining doping and cheating in e-cycling. Fincoeur and Bongiovanni [53] detailed how Zwift has evolved to be proactive against the would-be cheaters and/or dopers. They highlighted the potential challenges to tackling cheaters due to the new territory e-racing brings, how current anti-doping measures are entering a new territory to police this phenomenon, and how to track data and catch hackers. However, Thorne [54] expands on the issues of physiological monitoring in an effort to counter “weight-doping” and how this risks potential harm from excessive self-tracking and disordered eating. Riders [55] and journalists [56,57] have reported on the pressures of pre-race weigh-ins and the effect this can have on well-being, mental health, and physical performance.

We produced the first comprehensive and detailed review of the e-cycling anti-doping policy and outlined recommendations to strengthen and improve said policy [37]. The review discussed different methods of cheating, doping, or hacking a Zwift race. These include anthropometric manipulation (height and/or weight), gender doping, using PEDs, sandbagging (racing in a lower category), power manipulation (unusual pedalling styles) and lastly, power and controller manipulation (data fabrication and modification). Similar work by Dyer [58] grouped methods of cheating and doping in Zwift under four headings: avatar misrepresentation, in-game deception, data manipulation, and hardware manipulation.

Dyer [58] discussed the topic of cheating within e-cycling in the context of ethics, technology, and performance enhancements [59]. The work investigated “those who can participate in a competitive environment without fear of judgement or reprisal and in relative anonymity. Therefore, the level of temptation or incentive for athletes to cheat in e-racing is likely to increase as the potential rewards, such as prizes or their achievements, also increases” [59]. 

We [37] offered five recommendations to improve upon Zwift’s first-ever anti-doping policy, which pushed for dual recording for live events, removing doping tests at events not sanctioned by Zwift, creating a set of digital doping rule violations (DDRVs), linking with the court for the arbitration of sport (CAS) to strengthen anti-doping case hearings and relations with the IOC and the UCI, and Zwift fully adopting the WADA code. Lastly, Zwift’s anti-doping policy analysis also highlights that their current measurements do not promote clean sport and fail to protect athletes’ health and safety by their current anti-doping publication rulebook. Additionally, a “digital doping passport” [58], like the athlete’s biological passport (ABP) used by WADA [60] and a set of “digital doping rule violations” (DDRVs) [37], which would be the replication of the anti-doping rule violations as laid out by WADA, but specific and tailored for the competitive digital environments are needed. 

Zwift responded to our critique [37], stating: “Zwift as a platform is not competitive Esports-focused and the majority use the platform solely as a training aid. 80% will explore and free ride, 50% will train or complete a workout, and only 20% will compete. Of the 20% that race, only a small percentage is impacted by Zwift’s Cycling Esports Rulebook, either competing in the Zwift Racing League Premier Division or competitions like the UCI Cycling Esports World Championships. For our highest-level events, the World Championships, we have an extremely strong anti-doping program under full WADA rules and administered by the ITA/UCI. We are proud of the testing and rigour that goes into these events” [61]. 

This position is also apparent in many other esports games since they do not see themselves as all competition-focused, and some are purely there for player enjoyment, socializing, and community. We acknowledge that a complete anti-doping program is not appropriate for leisure users, and this distinction is essential to highlight.

Despite this, concerns of cheating and doping in Zwift and e-cycling are a concern for riders at all levels, which prompted us to create a survey to understand perceptions amongst riders.

## 5. Aims and Objectives

Aim: To assess if cyclists have experienced cheating during e-racing and in-person, and their perception of cheating and doping behaviours in online e-cycling.

Objectives: Are there differences in opinions between males and females around cheating, doping and anti-doping measures in e-cycling?Is Zwift’s statement correct that their perception of cheating is worse during in-person cycling compared to on their platform?Understand if the e-cycling community should adopt the same policies and punishments for cyclists as implemented in in-person events when they dope or cheat in e-races.Determine if e-cyclists are more at risk of engaging in performance-enhancing drug (PED) use than traditional cyclists.

## 6. Materials and Methods

### 6.1. Participant Information

Following approval from the Teesside University School of Health and Life Sciences ethics committee, an opportunity sample of 1467 participants was recruited via social networking sites (SNS) such as Facebook and Twitter and online forums such as Reddit. The criteria for inclusion to the study for participants had to be engaging in e-cycling and or cycling at any level. They would have to be over the age of 18 and be able to complete an online questionnaire. The survey was open for seven weeks in February and March 2021. Participant characteristics, including mean and standard deviation, are presented in Table 3 below.

Participants responded from 47 countries, with the United Kingdom (n = 718, 49%), the United States of America (n = 351, 24%), Canada (n = 88, 6%), Australia (n = 55, 3.8%), and the Republic of Ireland (n = 31, 2.1%) being the most common. 

### 6.2. Measures

The online survey was developed on Online Surveys.ac.uk (JISC, Bristol, UK) and designed by us to collect information on the participant’s e-cycling and cycling backgrounds and experiences, including (not limited to) anthropometric data, their training history, training methods, online training and racing, motivations for cycling, COVID-19, cheating, doping and anti-doping. The questionnaire used in the data collection was not of a validated design. It was created by the author team specific to the e-cycling community. The questionnaire was shared among cycling and e-cycling groups and forums across social media sites to obtain relevant information from the community we wish to analyse and understand. The methodology used is a cross-sectional descriptive study design, which is observational in nature. Participants received no monetary or external incentive to take part, all their personal information was anonymised and no identifying information was used. After reading the participant information sheet and instructions, participants gave written consent to the study. Questions were formatted as short answers, rating scales, and multiple-choice. Participants could opt out anytime during the study for any reason. The questionnaires took around 10–15 min to complete. 

### 6.3. Data Analysis

All responses were downloaded from Online Surveys into Microsoft Excel (v2016, Microsoft, Redmond, WA, USA) and Statistical Package for Social Science (SPSS) (V27, IBM, Armonk, NY, USA) for analysis. Questions on participants’ cycling experiences were coded and divided into sub-themes and reported as frequency plots, pie charts and tables across the topics of e-cycling and cycling. Descriptive statistics, including mean and standard deviation, were reported.

The data collection sample size was not decided by a specific number, only when data saturation had occurred [62]. Once enough participants had responded to the survey after reviewing their responses, we felt that the number of answers supplied by the diverse group of participants yielded no additional unique codes [63]. Furthermore, irrespective of male, female, age and nationality, the same instances were repeated over and over despite the diverse population of participants [63]. Hence, recruitment was closed after seven weeks to begin data analysis.

## 7. Results

The responses from participants for the nine questions related to cheating, doping, and policy for Zwift and e-cycling are presented below. All answers are rounded to the nearest whole percentage and displayed as a percentage of responses to that question (excluding “no comments” and irrelevant answers. Comparisons between males and females are not displayed in all the questions as there were no significant differences between these groups, which noted them to be reported in graphical format or written discussion below).


**“Have you ever experienced cheating during an in-person race?”**


From the sample of 1467 participants, 15% of participants answered “yes” to experiencing some form of cheating within an in-person race. Those who said “no” made up 44%, “no comment” made up 3% and 38% had not raced.


**“Have you ever experienced cheating during an e-race?”**


A total of 44% of participants selected “yes I have experienced some form of cheating within an e-race”, 40% selected “no” and those that had not e-raced were 4%; finally, 11% picked “unaware or don’t know”.


**“How did it make you feel when you were cheated?”**


Figure 1 summarises the responses to the question. A total of 51% were angry or annoyed at being cheated. However, total responses were low, partly due to the number of respondents stating they had not raced in the previous question.


**“Do you accurately input your height and weight for an e-race?”**


By sex, males answered 87% “yes”, 4% “no” and 8% did not make a comment. Females displayed a similar breakdown of results, with 91% responding “yes”, 4% responding “no”, and 6% did not make a comment.


**“Have you ever deliberately manipulated your height and weight for an e-race to gain an advantage?”**


95% of males and 96% of females reported they correctly input their details, and half of those who admitted altering their weight stated they purposely increased their weight to make racing harder.

**“Would you be more or less tempted to take a banned substance for e-racing?”** In Figure 2, both males and females reported low temptation to take banned substances for e-racing. Only 3% of male respondents and 2% of female respondents were more likely to take a banned substance despite the semi-anonymous nature of e-racing. Eleven per cent of total respondents were less tempted to take a banned substance. In total, 80% reported never feeling tempted to engage with PEDs. 


**“Do you think those caught cheating in e-racing should face a ban?”**


The responses to question 7 were varied, and participants went into different levels of depth with their answers. Coded responses are displayed in Table 4 below.

Responses were very similar between males and females; eighty-seven per cent felt there should be a ban to some extent or in some circumstances. Eight per cent said no; however, only males thought that those caught should be humiliated or shamed (1% of total male responses). 


**“Should that ban from e-racing also prevent them from taking part in in-person races as well?”**


Question 8 asked participants about the transfer of bans from e-racing to in-person racing. Opinions were split as to whether an e-cycling ban should also apply to in-person racing. Comments could be broadly condensed into “yes” (sometimes with additional comments), “no”, and “maybe” (depending on the infraction). The pie chart (Figure 3) displays the total responses to the question and again was very similar between males and females with little difference despite the significant population disparity by sex in the recruited sample.

**“Should that ban from e-racing also prevent them from taking part in in-person races as well?”** Respondents gave a wide array of answers to this question with different variations of yes and no, which can be seen in Table 5.

## 8. Discussion

Participants in this survey expressed their views on cheating and doping in e-cycling. The overwhelming majority do not want to see cheaters, dopers or hackers in their sports and online communities. Similar to their in-person cycling counterparts, they wish for the sport to be cleaner and fairer for participants to generate meaningful competition. However, in recent years, there is still a perception based on the members’ experiences, scientific literature and media that cheating and doping are not minor problems that will go away on their own. Cases have been reported by academics [36,53,54,58,64] and mainstream media outlets [44,65], which highlight this problem to the community. This may create the perception of increased risk-taking and cheating behaviours on the platform by members or observers of the events, for example, when the community reports on incidents that are widely shared across social media [66,67,68,69], although this may not factually be true. 

When comparing perceived cheating between in-person and e-cycling, it is obvious that participants experienced more cheating online, although it is likely due to the fact that it is easier to cheat online compared to traditional PED use, which enhances physical ability. This is likely also related to the issue of how many respondents compete in e-races compared to in-person racing. In our survey, 5% of males and 4% of females do not participate in e-racing, while 39% of males and 34% of females do not participate in in-person racing. In addition, due to the recent pandemic, more people have moved from outdoor racing to indoor racing on various platforms [18,29]. Therefore, it is more likely that they would have been aware of, seen, or heard of more online cheating. As mentioned above, it is considerably easier to cheat and manipulate an e-race compared to an in-person race, as reported in our article [36], which critiqued Zwift’s Anti-Doping Policy and listed a variety of ways one could cheat in e-racing. These include anthropometric manipulation (height and weight), gender doping, use of PEDs, sandbagging, power manipulation and power and controller manipulation. 

Our findings contradict Zwift’s statement [61] that there is less cheating online compared to in-person cycling. However, there are more total cases of cheating and doping in in-person cycling due to its long history. There is a large-scale anti-doping effort to clamp down on doping in the sport. Unfortunately, despite the investment from WADA, UCI and other anti-doping agencies, the level of cheating or doping is still high across the board for cycling and their respective disciplines for in-person events [70].

Despite the limited number of reported cases of e-racing cheating and doping, it perhaps could be viewed as a concern by e-athletes. Zwift has, in its relatively short existence, seen cases of cheaters and hackers looking to try and break the rules to their advantage [71,72]. Our results show that individuals have attempted to cheat from their own homes, where the scrutiny is much lower than in-person racing.

Participants who have experienced cheating expressed negative sentiments. Comments were divided into the following subcategories (excluding non-responses): 50% were “angry or annoyed”, 15% “didn’t care”, 11% “disappointed”, 9% felt “cheated”, 4% were “confused”, 3% felt it was “just part of racing”, 2% felt “doping is unacceptable”, and 1% felt “let down by the sport and the organisers”.

Most participants reported that they correctly input their height and weight for racing (96%). This is important, as the height of a cyclist affects how the avatar experiences aerodynamic drag in Zwift, and their weight will affect the speed, especially on inclines. Of those who did not correctly input height/weight, four participants reported they purportedly increased their weight to make racing more difficult. On the follow-up question, of 1467 responses, only four females and seven males admitted manipulating their height and weight for an advantage. What makes this interesting is the admittance of cheating in an online racing event via an anonymous survey and cheating through a virtual game doing a physically demanding activity. Ego or self-confidence is possibly the most likely reason for justifying their behaviour of cheating to win when altering their body shape to go faster, i.e., through the bends and uphill. 

The Zwift cheating survey [73] addressed the extent of height and weight manipulation in a similar survey of 600 participants. However, there may be a potential for bias towards Zwift as this was not peer-reviewed and was published on a website linked to Zwift. Regardless, the survey had some similarities to ours. For example, the average age ranges for men and women cyclists were very similar, with the majority being in their early forties, and participants were primarily from the UK and the USA. Finally, the breakdown by sex was very similar, with Robertson [73] reporting 80% male and 20% female vs. our 77% male and 23% female respondents. 

Notwithstanding, the cheating questions in their questionnaire were not as nuanced or specific as ours. Some of their results indicate a higher percentage of cheating than we report, but this may be due to the smaller sample size and because the questions did not delve into specific aspects of cheating and the impact of cheating or taking PEDs. However, they revealed a higher level of self-reported cheating; 46% of participants admitted to cheating at least once, and 7% admitted to cheating often [73]. The work also found that 12% of participants thought cheating on Zwift was common with their teammates or friends, while 60% felt cheating was common or common in general on Zwift. 

To understand the temptation to take a PED, we asked participants (on a varying scale of responses) to rate their likelihood of using a PED when racing. When excluding “no comments”, the vast majority said they were never tempted to take PEDs, with similar responses between males (80%) and females (81%). Six per cent of males and eight per cent of females were neither more nor less tempted; interestingly, 12% of males and 10% of females were less tempted to take PEDs for e-racing. In comparison, only 2% of males and females were more tempted despite the apparent lack of in-competition and out-of-competition testing from either ZADA or WADA at Zwift-sanctioned events where athletes compete remotely. 

When comparing sexes, female participants were more likely to report never being tempted to take a PED, while male participants were more likely to report being both “more tempted”, and “less tempted” for PED use for e-racing. This implies that males are more likely to cheat on Zwift. However, this does not mean females in cycling have never cheated, as there have been some examples. Although the number of high-profile cases is lower, it is most likely due to the total number of female professional cyclists being lower than male professional cyclists. One example of a female cyclist cheating is Katie Compton, a 15-time USA national cyclocross champion. She received a 4-year ban in 2020 after testing positive for an anabolic agent and did not contest the charges due to lack of money and subsequently retired [74]. It is not known whether she cheated throughout her entire career or only towards the end. 

There is no way to prove that there is a greater likelihood that someone will use PEDs and e-cycle compared to someone using PEDs and competing in an in-person cycling race. However, it would be easier to take PEDs at home if outside the Zwift testing pool of elite athletes on their platform, away from a WADA-tested event on the UCI professional cycling circuit. In addition to doping, there are other forms of cheating to elicit a performance advantage. There have been studies into cycling regarding the use of painkillers [75], Adderall [76] (a strong amphetamine stimulant), and the therapeutic use exemption (TUE) forms [77] to gain performance advantages by “legally” taking banned substances that without a TUE form would be classed as doping [78]. 

These responses reflect the literature investigating athletes’ perceptions and feelings around cheating, gamesmanship and/or doping when they have also experienced losing to someone who broke the rules to their advantage [79,80]. Hamilton and Coyle [79] describe the extensive doping practices in their book. They give a bleak overview of athletes feeling that doping was necessary to keep up and not waste a lifetime of training just to be ‘cheated’ out of winning by others who took PEDs. French cyclist Christophe Bassons exemplifies this; he possessed the same physiological characteristics (such as height, weight and VO_2_max) as Lance Armstrong, which, in theory, meant that he should have been able to compete at a similar level. Bassons believes that his decision to not dope cost him the opportunity to win almost any race and describes in detail the feeling of anger and hopelessness when not being able to keep up with doped contemporaries, especially after dedicating his life to the sport [80].

Over 75% of both males and females wanted some form of a ban for those who cheat in racing, only 5% did not want any form of ban or punishment, and around 15% of both males and females had no interest/no opinion/or did not care about this subject if people cheat or do not cheat in racing. This shows that most people feel a ban should be applied to cheaters and dopers in e-racing. However, the results differed when participants were asked if that ban should apply to in-person cycling as well. 

From those that provided an answer (excluding no comments), males and females had almost identical responses, with 56% of males and 57% of females being against an e-racing ban being applied to in-person racing, 34% of males and 34% of females felt the ban should carry over to in-person racing and 10% of both males and females answered “maybe”. This issue highlights that within our sample, there is a divide between traditional cycling and e-cycling regarding the potential implementation of bans across both cycling disciplines. 

Westmattelmann et al. [50] found that their sample of cyclists viewed punishments as effective and that increased bans deter doping. However, they stated that any extension of bans would have to be discussed on legal grounds as it could end up being a lifetime ban or, as the authors worded it, an “occupational ban”. The bans must be consistent across the sport and under WADA’s guidelines. The cyclists from this sample [50] also noted that fines are less effective and view the new anti-doping laws in their country as a valuable supplement to strengthening anti-doping policies. The previous questions and answers show that there should be a ban; the ban should be strong enough to deter athletes from cheating or doping, but extending the ban to other cycling disciplines is not a supported position. To understand this in more detail, the final question in our survey asked participants, “Should those caught cheating in e-racing face the same penalties as those caught cheating in in-person racing?”. 

From those providing an answer, 70% of males and 64% of females felt penalties should be the same, while 32% of males and 36% of females felt they should not. These results show that most males and females in our sample want the same penalties for those caught cheating or doping in e-cycling as they would receive in in-person cycling. However, these answers here conflict with the previous question regarding the ban crossing from one discipline of cycling to another. What these results have shown from our sample is that: (a) cyclists are in favour of a ban, (b) they are in favour of cleaning up the sport from dopers and cheaters, (c) they are happy to adopt bans and punishments from traditional cycling to e-cycling. Nevertheless, they disagree with a ban from e-cycling applying to in-person cycling due to the differences in the sports themselves, with many seeing e-cycling as a game or a training modality to help cycling and not seeing it as a competitive discipline of cycling. However, this question was towards the end of the entire questionnaire, so the number of “no comments” had increased, possibly due to fatigue or reduced concentration. 

This difference in opinions regarding the transferability of bans between modes of cycling shows that the UCI should consider the need to be more in touch with its membership base regarding fairness and rules. Creating events on Zwift and promoting e-cycling through the COVID-19 pandemic was helpful to cyclists. Similar to historical road cycling, they need to catch up when it comes to being proactive, preventing cheating and catching cheaters. 

This result also shows that traditional cyclists may see Zwift differently from in-person cycling as they might believe that it is not the real thing despite its immense benefit to their sport. However, research suggests that e-cycling and in-person cycling produce similar physiological markers and outputs when measuring the demands of their respective cycling disciplines [64]. Their results indicated that the physiological efforts of these races are comparable to in-person competitive cycling. In addition, e-cycling kept elite cyclists training and racing through COVID-19 restrictions and got more people interested in cycling with indoor training and smart trainers/smart bikes. Next, the UCI ran and broadcast online events on Zwift with professionals, enabling fans, elite athletes and cycling teams to interact and race together, a rare occurrence within the sport. From a participation outlook, indoor cycling is safer and a great entry point to cycling and racing that enabled many local clubs to create online teams to train together during COVID. Zwift and e-cycling are likely candidates to be the first esports at an Olympic game as it is a virtual game that requires physical effort to replicate a traditional sport, unlike most esports games, which are strategy and/or shooter-based, and therefore would not be considered. This positions Zwift in a unique position to be the best in both the physical and the digital world of esports. 

In June 2023, Singapore held an Olympic esports week which highlighted virtual sports [81]. With recent talk about the “Metaverse” hosting work meetings and events in a digital world where an individual could lead “another life”, Zwift could be the first metaverse sport, like in the movies ‘Ready Player One’ or ‘The Matrix’. It already has all the tools and equipment in physical hardware and games. The maps are created for cycling events and races, so transitioning over would not be complicated. Cyclists could wear a VR headset for a more “real-life cycling” experience. For example, a VR rider could cycle alongside the Tour de France riders and follow the feed/footage of the motorbike or bike-mounted cameras while trying to match the speed/power output of the cyclists. 

We posted the questionnaires to known e-cycling forums, group chats and websites with active online cycling communities. The survey did not specifically target competitive racers or in-person recreational cyclists so different results may have been generated for our sample compared to Westmattelmann et al. [50]. Furthermore, the survey was conducted at the start of 2021, which saw many countries worldwide still in lockdowns or working from home arrangements, which would have influenced many participants to cycle online. 

Finally, our study did not set out specific questions for cheating followed by specific questions for doping, as this would have made the questionnaire too long. Doing this would have likely revealed more in-depth answers to specific types of unfair play and gamesmanship within cycling and e-cycling. This is a potential limitation to our work, and for further studies in this topic area, adopting a validated scale such as the WADAs social science research package used by national anti-doping organizations would prove to be a useful tool for validating any results [82]. 

It is important to state that the difference in our sample between those who have and have not competed in e-racing may have created a self-reporting bias. With the number of recorded incidents of cheating in e-cycling and our questionnaire being published, this may have given the opportunity for some participants to potentially exaggerate their responses, especially if they have been cheated out of winning an event. Secondly, those who won an e-race who cheated may not want to admit they did and alter their results. Furthermore, for those respondents who had not e-raced, their opinions would have affected the results as they do not have any experience, and this may also explain their views as many respondents felt e-cycling is a training aid or a game rather than its own cycling discipline. Finally, we collected our data using social media forums and groups which actively participated in e-cycling and or cycling, which may have created a self-reporting bias. However, participants volunteered to take part, there were no monetary incentives, and our study received no funding to be carried out. For future research, a validated questionnaire should be employed to help increase the validity and reliability of responses from this population group when discussing cheating and doping. 

Nevertheless, the work here has produced exciting findings for e-cycling organisations and cyclists. From the sample recruited, the Zwift community have a range of opinions based on their experiences of cheating, being cheated, and agreeing upon the setting of anti-doping and e-cycling sporting policies. This is despite Zwift emphasising that their platform is mainly for recreational cyclists and has a lower rate of cheating than traditional cycling. Regardless, many members are unhappy with experiencing and seeing cheating taking place, as reported in this paper and other research literature [53,54,58,64]. 

Since our publication [36], which called for an update of Zwift’s anti-doping policy alongside the creation of the DDRVs, there has been an increase in instances of digital doping or hacking on Zwift without any update from this original policy and evidence of a comprehensive plan to tackle and reduce the hackers on the platform. 

Interestingly, the results were divided when the authors asked the community what they would like to do regarding cheaters. These differences appear at their most significant when transferring bans from the digital realm (e-cycling platforms) to reality (in-person racing). This is due to many of the responses feeling that e-cycling is a game or an addition to their training, which should be placed on a different importance level than in-person competitive cycling. However, when asked, the overwhelming majority of respondents wanted a ban to be enforced for any level of cheating when using an e-cycling platform and a ban that significantly impacts the perpetrator. More work is needed to fully understand if traditional cyclists want their bans to transfer to virtual cycling and if they would welcome a robust ruleset from Zwift that also applies to their in-person competitive calendar.

## 9. Conclusions and Policy Implications

The analysed respondents from our questionnaire leaned towards wanting bans for rule violations in e-cycling; however, they do not necessarily want them to carry over to in-person cycling. This is due to traditional cyclists’ comments in this study viewing e-cycling as either a game or a training mode. Many respondents’ answers wanted the same punishments for in-person cycling to be applied to e-cycling. E-cycling may be seen as a game or a mode of training, but participants in our study do not want cheaters or dopers present in their virtual racing environments as they see it as a strict mode of competition. Finally, in this study, there were few differences between male and female participants in their views on doping and cheating in e-cycling or cycling.

The work here should be considered in combination with our anti-doping paper [36]. Given the large sample size representing the community perspectives, Zwift and other e-cycling platforms may use our work as an example of how e-cycling members view their experiences of being cheated and what appropriate action should be taken against those who cheat and dope.

In the aftermath of the 2022 Birmingham Commonwealth Games and the removal of the esports events at the next games in Australia due to the issues around cheating and doping, traditional esports and sports with virtual competitive events must be proactive in tackling and removing any digital dopers and cheaters from their gaming environment [83,84] if they wish to be considered for world championships, Commonwealth games and the Olympic games.

For policymakers and e-cycling platforms such as Zwift, the potential implications for these findings are supported by the new direction Zwift is taking for their esports cycling events. In 2021, Zwift stated that the competitive racing side of their platform makes up 20% of their entire player base. However, in recent months, this is growing at an increasing competitive rate of 10–20% monthly. Furthermore, Zwift has said they now have 2000 registered teams for the 2023 indoor season, which will be a new record for the up-and-coming season [85].

In addition, Zwift and indieVelo have partnered to further legitimise cycling esports with greater oversight, independent oversight and improving top-tier cycling [86]. Investigating some of the methods to prevent cheating and hacking, indieVelo stated, “records your smart trainer and power meter into a single dual-recorded FIT file in-game to allow for easy one-step dual recording” [86,87,88], which was one of the original suggestions by Richardson, Smith and Berger [36] to push for a dual recording system to prevent a digital doping rule violation (DDRV). As e-cycling grows, the International Olympic Committee (IOC) is hosting the first-ever esports series in Singapore [81] and the continued investment into future esports events is evident with six cities interested in hosting the 2024 Olympic Esports week [89]. Taking all of this into account for the sport of cycling, Zwift is hailed as the representative platform for cycling and is one of the more popular esports games at this event. Recently, the UCI has hinted for an international calendar and ranking system for e-cycling events alongside a three-tiered approach for hardware, performance and technology standardisation [90]. This coincides with a new doping case in cycling where an athlete who competes in both in-person and e-cycling was caught using an anabolic steroid when tested during an in-person race [91]. Ultimately, the need to eliminate would-be hackers, cheaters and dopers grows even greater, especially with the importance of clean sport during the Olympic esports week.

## Figures and Tables

**Figure 1 sports-11-00201-f001:**
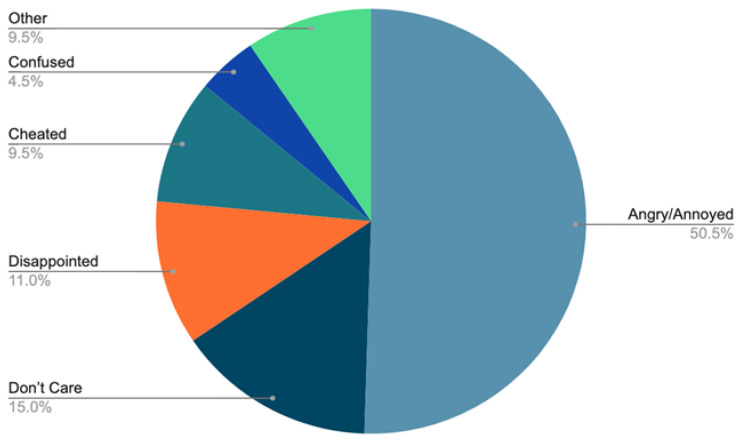
Participants’ opinions on being cheated.

**Figure 2 sports-11-00201-f002:**
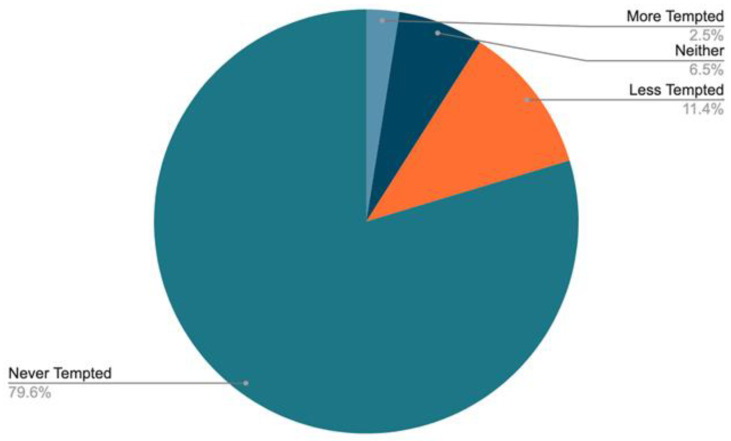
Participants’ temptation to take a banned substance.

**Figure 3 sports-11-00201-f003:**
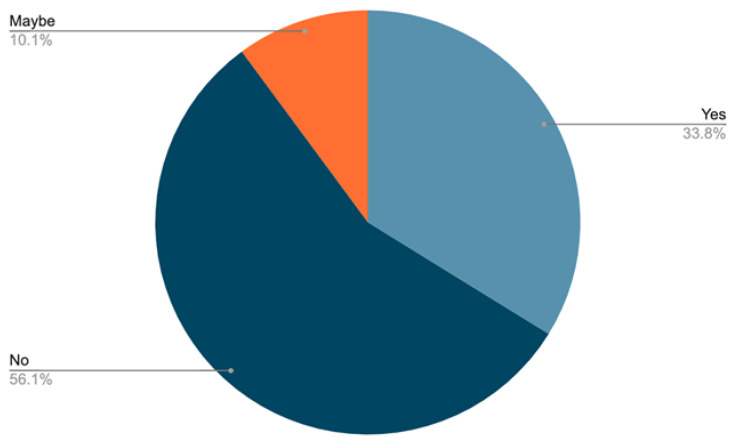
Participants’ opinions on bans in e-racing applying to in-person racing.

**Table 1 sports-11-00201-t001:** The total prize purse for selected esports and traditional sports [13,14,15,16].

Event	Total Prize Purse
League of Legends (2018)	$6,450,000
Arena of Valour (2022)	$10,000,000
Fortnite (2019)	$15,287,500
DOTA 2	$40,018,400
Australian Open Tennis (2023)	$76,500,000
UEFA Champions League (2022)	$2,032,000,000
Formula 1 (2023)	$2,200,000,000

**Table 2 sports-11-00201-t002:** The number of cyclists racing at each level pre- and post-COVID-19 [37].

Year	UCI World Tour		UCI ProTeam
	Males	Females	Male
2019	487	593	487
2023	531	231	405

**Table 3 sports-11-00201-t003:** Participant characteristics.

	Total	Males	Females
Number (n)	1467	1130	337
Age (Years)	46 ± 11	46.5 ± 11.4	54.4 ± 10.8
Cycling Training History (Years)	21 ± 16	22.4 ± 16	16 ± 14

**Table 4 sports-11-00201-t004:** Participants’ opinions on e-racing bans.

Response	Percentage of Responses	
	Males (%/N)	Females (%/N)
Yes- no example	57% (640)	54% (182)
Yes- lifetime ban	1.2% (13)	0.9% (3)
Yes- if financial gain	1.7% (19)	3% (10)
Yes- with some first warning, then a ban for repeat offenders	0.3% (3)	0.3% (1)
Yes- a month or temporary ban	2.2% (25)	3.9% (13)
Yes- depends on the race level for the severity of the ban	4.2% (48)	3.9% (13)
Yes- at higher level races (pros)	5.0% (56)	5.6% (19)
Yes- but how do you police it?	1.3% (15)	4.5% (15)
Yes- humiliate/shame them	1.0% (11)	0.6% (2)
Yes/No	0.3% (4)	0.0% (0)
No- no example	5.3% (60)	0.3% (1)
No comment	8.0% (90)	3.9% (13)
Not sure/do not know	2.8% (32)	9.5% (32)
Do not care/not interested	1.1% (12)	4.5% (15)
No- bans will not work	0.2% (2)	0.9% (3)
No- it is not real cycling it is a game	1.5% (17)	1.5% (5)
Other comments	1.6% (18)	2.4% (8)
**Total**	100%/1130	100%/337

**Table 5 sports-11-00201-t005:** Participants’ opinions on penalties in e-racing compared to in-person racing.

Response	Percentage of Responses	
	Males	Females
Yes- cheating is cheating	2.7% (30)	2.4% (8)
Yes- if financial gain	1.8% (20)	1.5% (5)
Yes- if doping or drugs are used	1% (11)	0.3% (1)
Yes- no example	43% (484)	36% (121)
Yes- temporary ban	0.4% (4)	0.0% (0)
Yes- if governed by the UCI or formal body	0.5% (6)	1.8% (6)
Yes- depends on race level for the severity of the ban	5.4% (61)	3.9% (13)
Yes- at higher race levels (pros)	2.4% (27)	4.5% (15)
How would these rules/bans be comparable or enforced?	1% (11)	1.2% (4)
I do not know	2.5% (28)	2.1% (7)
Maybe	0.9% (10)	1.5% (5)
No- no example	24% (275)	28% (94)
No- it is just a game	0.5% (6)	0.9% (3)
No opinion	1% (11)	0.9% (3)
Not sure	1.6% (18)	1.5% (5)
No comment or N/A	11% (125)	13% (45)
Other comments	0.3% (3)	0.6% (2)
**Total**	100%/1130	100%/337

## Data Availability

The data presented in this study are available on request from the corresponding author. The data is not publicly available due to ethical restrictions to protect the identity and privacy of participants.

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
