# Peer review of "Perceptions of Cheating and Doping in E-Cycling"

_sports, 2023, doi:10.3390/sports11100201_

Round 1

Reviewer 1 Report

E-sport is developing dynamically and, like traditional sport, it carries various risks. One of them is doping. While in the case of traditional sport this phenomenon has been recognized and studied for a long time, doping in e-sport is something new that needs to be known. For this reason, I consider the research contained in the article entitled "Perceptions of Cheating and Doping in E-Cycling" to be important and innovative. Research in this area seems necessary to be able to undertake effective anti-doping activities in the future. Especially since the problem is probably bigger than it seems. The shocking fact, resulting from the research described in the article, is that as many as 44% of respondents experienced cheating during e-races. It should also be borne in mind that in the digital world doping can have a completely different dimension than in real life.

Overall, the article is well written, clear and understandable, and the content is interesting. So my rating is positive. However, I do have a few comments.

Introduction and theoretical part

In this part of the work, the authors discuss in detail the issues of e-cycling, as well as cheating, doping and anti-doping activities in cycling. They also characterized doping and cheating in e-sports and e-cycling. While detailed information on e-cycling is justified, writing about doping in sport seems unnecessary, because the information on this subject is widely known, so I suggest shortening this part of the text, which is very extensive.

Materials and Methods

The number of people tested is sufficient, however, there is a lack of information on the criteria for admission to research and rejection. I suggest supplementing this information.

Measures

There is no information whether the questionnaire used in the research has been subject to any form of validation.

Data Analysis

The subsection number (6.1.) for "Measures" is the same as for "Data Analysis", it should be changed.

In addition to descriptive statistics, some static methods, such as Chi-square, should be used.

Why were only three pie charts with user responses used, even though there were more questions?

Figure 2 and none of the tables have been referred to in the text, this should be supplemented. This also applies to tables in other chapters.

In addition to the percentages, the tables should also include the number (n) of people who answered the question.

Author Response

Thank you for your comments and suggestions

Kindest regards

Reviewer 2 Report

Introduction

The introduction provides a comprehensive overview of the evolution and significance of esports and its connection to traditional sports, specifically in the context of motion-based video games. However, it might benefit from a more focused and concise approach to better highlight the main themes of the paper.

The introduction briefly mentions the rapid growth of esports and lists popular games without delving into the specific factors contributing to their popularity or the nature of their dedicated leagues and tournaments. Expanding on these aspects with more context and detailed explanations would provide a richer understanding of the esports landscape.

The introduction should provide clearer transition sentences between different topics, such as the shift from discussing esports to e-cycling and the focus on cheating and doping issues.

Methods

To ensure transparency and establish the representativeness of the sample, consider providing information about any prior sample size calculations or considerations considered when determining the target number of participants.

In addition to describing the measures included, it's advisable to briefly address the validity of these measures. Highlighting any efforts taken to establish the reliability and validity of the survey questions will enhance the credibility of the collected data.

Results

In relation to Figure 1, while the visualization of participants' feelings when cheated provides insights, it's crucial to acknowledge the potential bias due to a subset of respondents not having raced.

The graphical representation of opinions, as shown in Figures 1, 2, and 3, aids in visualizing the responses. It is valuable that opinions are represented by both males and females, highlighting potential gender-based differences in views.

Discussion

While it's mentioned that participants experienced more cheating online compared to in-person events, it would be beneficial to explore the potential reasons behind this observation.

Addressing counterarguments and limitations can demonstrate the depth of analysis and the thoroughness of your study. Additionally, discussing the potential limitations of the survey itself, such as potential biases or self-reporting inaccuracies, would further contribute to the transparency and credibility of the research.

This section could also benefit from a section that outlines potential implications for policy and future directions. How could the insights gained from this survey influence anti-doping measures, policy formation, or educational initiatives within e-cycling platforms?

Author Response

(The authors gave the same response as above.)

Reviewer 3 Report

This is a very well-written manuscript. Below are my comments.

The introduction and literature review are well written with clearly defined aims.

Methodology:

Can you please elaborate on the questionnaires used. Perhaps put the questionnaire in a supplementary section and refer to it. 

Were any of questions used from validated questionnaires?

Did any of the questions get validated to determine that they were capturing what you think they are capturing? 

Further, did any of the questions allow for multiple responses (e.g., "How did it make you feel when you were cheated?"). Someone can feel multiple things simultaneously. 

How did you account for social desirability bias in your questionnaire?

You should also identify in your methodology that this is a cross-sectional descriptive study. 

Discussion

Based on the data that you have collected I think you make several jumps in your discussion section, especially since many of these findings could be impacted by social desireability bias which your questions did not really try to control for. 

Author Response

Please see attachment document

Thank you for your suggested comments

Kindest regards

Round 2

Reviewer 2 Report

The authors did a good job in reviewing their manuscript.

Author Response

We would like to thank the reviewer for accepting our revisions to our manuscript. 

Thank you for your efforts in helping make our paper better.

Kindest regards

Reviewer 3 Report

Thank you for responding to my concerns. Your responses have raised additional concerns.

1. How did you check for central limit theorem? In a qualitative study you can look at themes whereas in a quantitative study the CLT states that the data is remaining normally distributed. However based on the data that you have presented in your results your data is not normally distributed. Further, many of your questions are categorical in their responses, and you may make an argument that they are ordinal however, even ordinal data cannot be plotted using CLT.

2. I still don't see a way that you controlled for social bias. You did not include any test questions in there and the questionnaires are not validated. Many of your answers could simply be a result of social desireability bias.

3. In your discussion section you should tone down your language to use words such as may because this study design and questionnaire does not lend itself to making conclusive results, despite the large data set.

Author Response

Please see attached word document regarding your comments. We would like to thank you for your work. 
